

# Investigation of a low-cost magneto-inductive magnetometer for space science applications

Leonardo H. Regoli[1], Mark B. Moldwin[1], Matthew Pellioni[1], Bret Bronner[2], Kelsey Hite[1], Arie Sheinker[3], and Brandon M. Ponder[4]

[1]Climate and Space Sciences and Engineering, College of Engineering, University of Michigan, Ann Arbor, USA
[2]Space Physics Research Laboratory, College of Engineering, University of Michigan, Ann Arbor, USA
[3]Magnetic Sensing, Soreq Nuclear Research Center, Israel
[4]Nissan Technical Center North America (NTCNA), USA

*Correspondence to:* Leonardo H. Regoli (lregoli@umich.edu)

**Abstract.** A new sensor for measuring low-amplitude magnetic fields that is ideal for small spacecraft is presented. The novel measurement principle enables the fabrication of a low-cost sensor with low power consumption and with measuring capabilities that are comparable to recent developments for CubeSat applications. The current magnetometer, a software-modified version of a commercial sensor, is capable of detecting fields with amplitudes as low as $8.7\ nT$ at $40\ Hz$ and $2.7\ nT$

at $1\ Hz$, with a noise floor of $500\ pT/\sqrt{Hz}$ @ $1\ Hz$. The sensor has a linear response to less than $3\%$ over a range of $\pm 100,000\ nT$. All of these features make the magneto-inductive principle a promising technology for the development of magnetic sensors for both space-borne and ground-based applications to study geomagnetic activity.

## 1   Introduction

Magnetic fields are a ubiquitous feature of our solar system and of key importance for geophysical, magnetospheric and

heliospheric investigations. The sun produces the interplanetary magnetic field (IMF) and many of the planets and moons throughout the solar system produce their own magnetic fields through dynamo and magneto-inductive response processes. Even where no internally produced magnetic field is present, for example, Mars, or Venus, the IMF plays a major role in how planets and smaller bodies interact with the solar wind.

At Earth, the measured field is a combination of the internal dynamo-generated field and perturbations that occur in space,

particularly during substorm and geomagnetic storm processes. These processes are governed by the direction of the IMF and the dynamic pressure exerted by the solar wind at any given time (e.g., Moldwin, 2008). The enhancement of the particle fluxes in the ring current during a geomagnetic storm causes the measured magnetic field strength at the surface of the Earth to decrease. This is quantified by the so-called disturbance storm time (Dst) index, which is determined by a network of low-latitude magnetometers (e.g., Hamilton et al., 1988; Liemohn et al., 2001).

The dynamic nature of planetary magnetospheres makes it extremely difficult, if not impossible, to understand their structure without the help of a magnetometer with sufficient resolution, dynamic range, and bandwidth, to discriminate between the different regions inside the magnetosphere and identify the magnetic signature of plasma flows that are governed by global and

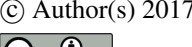



local circulation patterns. For this reason, magnetometers have been a key tool in magnetospheric investigations throughout the history of their study and continue to be indispensable. Critically, current and planned investigations of multi-scale dynamic features throughout the solar system continue to drive the need for greater numbers of magnetometers with state of the art capabilities.

## 1.1 ULF waves in the magnetosphere

The Earth's magnetosphere, whose field strength varies from about $60,000\ nT$ in polar LEO orbit to about $100\ nT$ at geosynchronous orbit, have different wave populations present with frequencies ranging from a few mHz to a few Hz on both the day- and nightside. Traditionally, the continuous pulsations which are denoted by Pc1-5 can be divided into categories that are characterized by a given frequency range as summarized in Table 1 (e.g., Jacobs et al., 1964; Fraser, 2007; Menk, 2011).

**Table 1.** ULF waves in the magnetosphere.

| Wave | Frequency |
|------|-----------|
| Pc1 | $0.2 - 5\ Hz$ |
| Pc2 | $0.1 - 0.2\ Hz$ |
| Pc3 | $22 - 100\ mHz$ |
| Pc4 | $7 - 22\ mHz$ |
| Pc5 | $1 - 7\ mHz$ |

These waves provide an insight into magnetospheric dynamics including wave-particle interactions (mostly Pc1 and Pc2), the solar wind activity (Pc3 to Pc5, Takahashi et al. 1984, Takahashi and Ukhorskiy 2008), as well as internal processes (e.g., Hartinger et al., 2014).

Several studies have focused on the relationship between fluctuations in solar wind conditions and the observation of ULF waves in the magnetosphere and on ground stations. Among others, Kessel (2008) used data from the ACE, Wind, Geotail, Cluster and GOES satellites and from ground stations to perform a statistical study during a period of time of over a month and found that for most of the time when Pc5 waves were observed, their amplitude and power were related to fluctuations in the solar wind, with only about 20% of the total power coming from internal processes. A similar dependence on solar wind conditions has been observed for waves in the Pc3 (e.g., Constantinescu et al., 2007; Clausen et al., 2009) and the Pc3-4 range (Heilig et al., 2007).

Given their dependence on different aspects of the interaction between the solar wind and the magnetosphere, Pc3 to Pc5 waves, when measured on the dayside, provide a way of studying how the global magnetosphere reacts to changes in heliospheric parameters such as solar wind density, solar wind speed and IMF (e.g., Shen et al., 2015; De Lauretis et al., 2016; Takahashi et al., 2016; Shen et al., 2017).

The use of ground-based magnetometers, depending on their distribution around the globe and in combination with global models of the magnetosphere, can also shed light on how global the distrubances are, by correlating the signals observed at different latitudes with the length of the corresponding magnetic field lines.



In addition, and due to the dependence of the Alfvén velocity on the local plasma denstiy, ground magnetometers can be used to infer low-energy populations that are difficult to measure in space due to spacecraft charging effects (Menk et al., 1999). In a similar manner, the observation of field line resonances has been used to infer other properties of the magnetosphere such as location (Dent et al., 2006), density (Berube et al., 2003) and composition (Takahashi et al., 2008) of the plasmapause or also the location of the open-closed field line boundary (Ables and Fraser, 2005).

One of the difficulties of studying waves in the magnetosphere is that conditions change rapidly and thus standing waves are difficult to maintain (Kivelson, 2006). This translates into a strong damping of the waves and thus multi-point observations are necessary to study the different regions affected at the same time.

The use of small satellites with commercial off-the-shelf (COTS) instruments on board opens the possibility of having large, cost-effective constellations, making it possible to study both large structures that are visible at magnetosphere-scales as well as small structures such as magnetic reconnection that are close to the electron scales (Burch et al., 2016).

## 1.2 Measurement approaches

With the growing interest of the scientific community in small satellites as a tool to perform magnetospheric and heliospheric studies, the need for space instruments that are cheaper and easier to produce has given rise to the study of different possibilities including the use of COTS components or complete instruments. Originally CubeSats were mostly seen as technology demonstration platforms, however there are now missions with scientific instrumentation being flown and proposed (e.g., Moretto, 2008; Springmann et al., 2012; Klesh et al., 2013; Heine et al., 2015; Lepri et al., 2017; Goel et al., 2017; Zurbuchen et al., 2016).

When it comes to magnetometers, due to their reliability, performance and ability to measure low fields, two types of sensors have predominantly been used for space missions, namely fluxgate and helium magnetometers. However, due to their high fabrication costs, relatively large size and high power needs, different alternatives have been recently studied for CubeSat missions. One approach is to miniaturize fluxgate magnetometers, while the other is to explore chip-based COTS technologies such as magneto-resistive and Hall magnetometers.

Miles et al. (2016) developed a miniature fluxgate magnetometer with noise floor of about $200\ pT/\sqrt{Hz}$ @ $1\ Hz$ with a power consumption of $400\ mW$, considerably smaller than instruments used in large missions such as Cassini (up to 12.63 $W$ combining a fluxgate and a vector helium magnetometer, Dougherty et al. 2004) and, more recently, the Magnetospheric Multiscale Mission (MMS, almost 2 $W$ for the fluxgate instrument, Russell et al. 2016).

Also using the fluxgate measurement principle, Matandirotya et al. (2013) compared three COTS instruments with special focus on parameters relevant for space applications. They identified one specific magnetometer, the LEMI-011B that, after modifications involving separation of the sensor and the electronics, was able to achieve noise levels below $1\ nT$ @ $12.83\ Hz$ and 30 mW power consumption.

Taking advantage of the mobile phone popularization that happened during the last decade, Ponder et al. (2016) evaluated the possibility of using a smartphone magnetometer to perform magnetospheric studies, finding that, despite the relatively poor performance of these magnetometers in terms of resolution, they could still be used to detect certain processes happening



at different regions of the magnetosphere such as field-aligned currents (FAC), auroral electrojet and equatorial electrojet signatures.

A similar approach was taken by the AMPERE project (Anderson et al., 2000) making use of the engineering magnetometers on-board the commercial satellites of the IRIDIUM constellation. With a resolution of 48 nT, they are able to capture signals

above the noise level of the instruments at high latitude that correspond to perturbations coming from FACs. The availability of a large number of satellites (>70) makes it possible to obtain global measurements of Birkeland currents with a time resolution impossible to obtain with smaller configurations.

Using a hybrid approach, Brown et al. (2014) achieved sensitivities below $3\ nT$ with the use of magnetoresistive technology. The power consumption of their instrument is comparable to that achieved by the one presented by Miles et al. (2016), at 425

$mW$ for science applications. Their instrument also provides the possibility of being used for attitude determination. Under this latter mode, the power consumption is reduced to $140\ mW$.

Using the anisotropic magnetoresistance (AMR) measurement principle, Fish et al. (2014) developed a sensor specific for the DICE CubeSat mission. They achieved a resolution of $5.5\ nT$ over a range of $1500\ nT$. The sensor also includes a degauss circuit to counteract saturation of the AMR elements. In total, the sensor has a weight of $25\ g$ and a power consumption of

$108.9\ mW$. AMR magnetometers are, however, limited by the dependence of their gain on the magnitude of the field. This sets a strong limitation when trying to measure low fields.

A comprehensive review of commercial magnetometers suitable for space applications until the end of the previous decade, including a historical perspective, can be found in Diaz-Michelena (2009). In general, small sensors present important progress in terms of size, mass, power consumption and cost and, while the measurement capabilities are still not at the level of those

developed for larger missions, the resolutions and noise levels are already at values sufficiently good to be considered for small scientific missions.

In this paper we present the testing and characterization of a software-modified version of a COTS magnetometer in the scope of reduced size, weight, power and cost (SWAP+C) space physics applications. The RM3100 magnetometer, built by PNI Sensor Corporation, is based on a measurement principle significantly different from the current standards for space

applications like fluxgate, helium, magneto-resistive or Hall magnetometers.

The sensor is intended for Earth-based applications and is particularly well suited for automotive applications such as compassing or for detection of nearby objects due to its small size. However, its performance under magnetic field conditions observed at planetary magnetospheres, as evaluated in this study, makes the magneto-inductive technology a promising one for low-cost space missions based on small satellite technologies. Taking into account the determined resolution and frequency

response of the instrument, its performance is promising for the study of magnetospheric ultra-low frequency (ULF) waves and small-scale current systems.

## 2 The magneto-inductive technology

The RM3100 magnetometer (shown in Figure 1) is based on the magneto-inductive principle. All the data presented in this paper were gathered with a modified version of the COTS sensor. The modifications thus far consist of optimization of internal parameters of the sensor that allowed us to improve its performance. These are software modifications and no hardware changes

were made for any of the tests described.

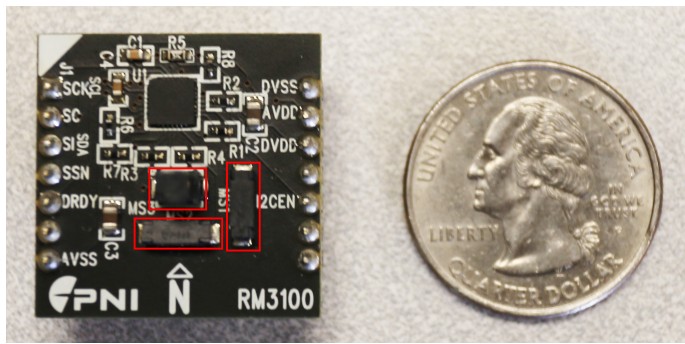

**Figure 1.** PNI RM3100 magnetometer shown next to a quarter coin for size comparison.

The COTS version shown in Figure 1 consists of the orthogonal coils (indicated with red rectangles), and an ASIC controller with the peripheral electronics. In order to communicate with the ASIC controller, a PNI CommBoard was used. The CommBoard introduces a constant interference of up to several hundreds of nT, which is easily eliminated by removing the bias. However, this CommBoard is only used for test purposes. A new dedicated electronic circuit and a microcontroller has

been designed for integration into a small satellite.

The sensor is a simple RL circuit that does not use an A/D converter, one of the electronic components that is sensitive to external radiation in traditional fluxgate magnetometer designs. The operating principle of the PNI sensor involves measurement of the time it takes to charge and discharge an inductor between an upper and lower threshold by means of a Schmitt trigger oscillator. This time is proportional to the applied field strength, within a specified operational range. A simple diagram of the

electronics is shown in Figure 2, taken from Leuzinger and Taylor (2010).

The total field that the sensor experiences is due to the external field and the field generated by the circuit ($H = kI + H_E$, where $k$ is a property of the coil, $I$ is the current through the circuit and $H_E$ is the external field). The Schmitt trigger causes the current through the circuit to oscillate as the voltage across the resistor ($R_b$) passes a set 'trigger' value. As the applied current oscillates the inductance of the circuit (and hence the time constant) changes. This behavior can be seen in Figure 3.

An applied magnetic field causes a constant offset in the coils' field strength, the polarity of which is determined by the direction of the field. This offset causes the average permeability and therefore inductance to be lower in one direction and larger in the other yielding a corresponding difference in the time required to complete the minor B-H loops in each direction. By integrating over many such minor loops in each direction, the time difference, and therefore available resolution, can be enhanced to any desired level subject to integrated noise sources.





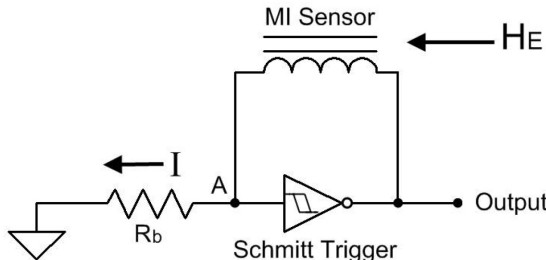

**Figure 2.** Schematics of the electronics involved in the magneto-inductive technology (from Leuzinger and Taylor (2010)).

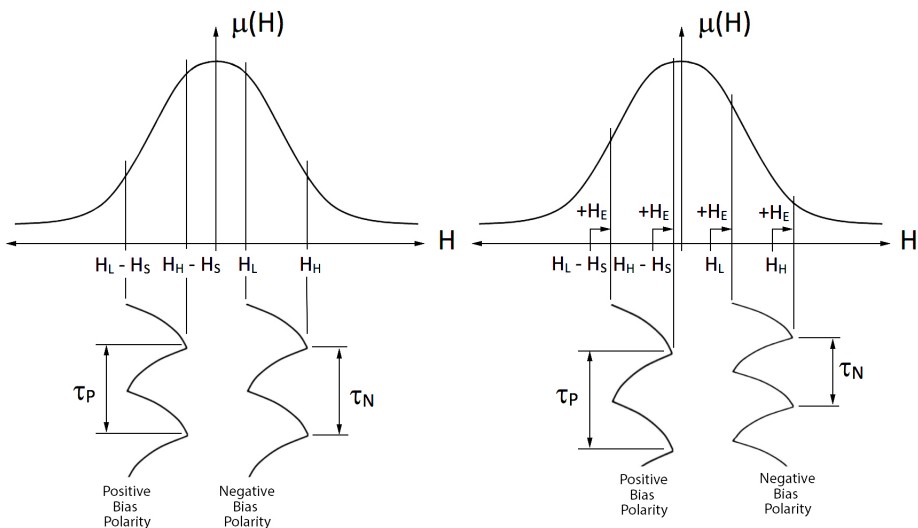

**Figure 3.** The induction in the coils as a function of applied magnetic field (top) and the traces of the oscillating current in the solenoid and the period for positive and negative bias polarity (bottom) (from Leuzinger and Taylor (2010)).

In the commercial version of the sensor, the number of loops used for integration is controlled by a register called cycle count and, for all the experiments presented in this paper, this value was set to 800. The value of this register is inversely proportional to the sampling frequency which, for this paper was of approximately $40\ Hz$.

The lack of an A/D converter, together with the absence of amplifiers required in traditional fluxgate magnetometers reduces the power consumption, the mass and the size. Additionally, being a COTS chip-based sensor, it is possible to produce it in large batches, something much more difficult in the case of larger sensors such as fluxgate or helium magnetometers.




In addition to the described advantages, a similar sensor based on the same technology and also built by PNI, the MicroMag3 (basically an earlier version of the RM3100), has been flown in space on-board a CubeSat (Springmann et al., 2012). The sensor, part of the RAX mission, was used as part of the attitude determination system.

## 3    Characterization of the sensor

Five different tests were performed to characterize the sensor. The tests were aimed specifically at determining the sensor's resolution (defined as the root-mean-square or RMS noise), stability, linearity and frequency response. In the beginning of this section, the testing facilities used during the data collection are described. Then, the tests performed will be presented, along with their corresponding results.

For all the tests presented in this paper, a sampling frequency of $40\ Hz$ was used while taking individual measurements for

each of the axes which are either analyzed separately (like in the case of linearity) or together in the form of field magnitude (for the rest of the tests). The orthogonality of the coils was not studied but will be the focuse of future tests.

The tests were carried out using a three-layer shield can and a copper room (with $\mu$-metal lining), shown in Figure 4, in order to reduce the influence of the Earth's magnetic field and any additional interference during the data collection. Both structures are available at the Department of Climate and Space Sciences and Engineering (CLASP) and the Space Physics Research

Laboratory (SPRL) at the University of Michigan.

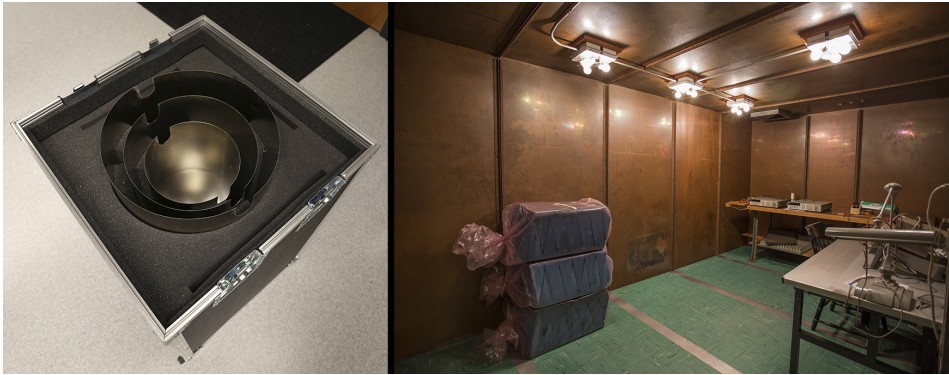

**Figure 4.** Three-layer shield can (left) and copper room (right) used for the characterization for the PNI RM3100 magnetometer.

In order to assess the ability of the shield can and the copper room to isolate the external field, 30 second measurements without applying an external field were taken with different configurations, either inside or outside the shield can and the copper room in every possible combination. This test also served to measure the resolution of the RM3100, so the results are presented in the relevant sub-section.

As a cross-calibration tool to contrast the values obtained by the RM3100 magnetometer, a Meda uMAG fluxgate magnetometer with a $1\ nT$ resolution was used (MEDA, 2005). This resolution is about an order of magnitude finer than that of the unmodified RM3100 according to the specifications provided by its manufacturer.



For all the tests presented in this section, the sensor was run at a sampling frequency of 40 Hz. While the sensor can be run to much higher frequencies, the choice of 40 Hz, which would allow measurement of periodic signals with frequencies up to 20 Hz, is based on the applicability of the magnetometer to measure ULF waves in the magnetosphere.

## 3.1 Resolution

5   The resolution of the sensor represents the minimum change in the strength of the applied magnetic field that can be detected. Its value is directly related to the the least significant bit of the sensor and its inherent noise floor, which must be taken into consideration during testing. As stated before, the resolution was determined using the shield can and the copper room at CLASP and SPRL. The configurations are listed in Table 2.

**Table 2.** Resolution tests set-up and results.

| Configuration | Description | STD |
|:---:|:---:|:---:|
| 1 | Outside shield can, outside copper room | 34.23 |
| 2 | Inside shield can, outside copper room | 11.26 |
| 3 | Outside shield can, inside copper room | 8.84 |
| 4 | Inside shield can, inside copper room | 8.73 |

The results are shown in Figure 5, where the de-trended measured magnitude of the magnetic field is presented. In the
10   legend, SC stands for shield can and CR stands for copper room.

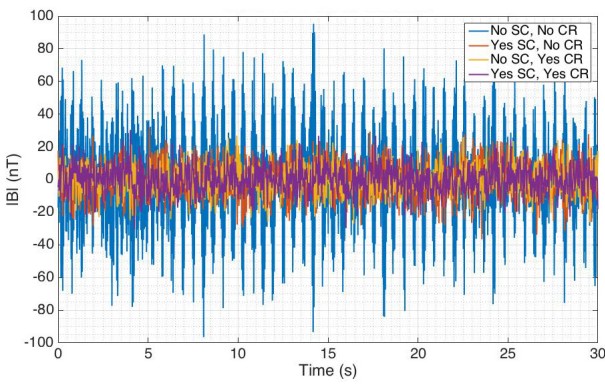

**Figure 5.** Zero applied field measurements under different shield can/copper room configurations.

Since no external field is being applied during the data collection, any field present will correspond to a combination of the Earth's field and the one produced by AC power lines (60 Hz) which is the primary cause of the large oscillations present in the blue curve. The 60 Hz signal is still present when the sensor is placed inside the shield can but outside the copper room



(configuration 2) but it is efficiently removed for configurations 3 and 4. The Earth's field is removed from the curves by de-trending the data before producing the plots.

From the standard deviation of the measurements (STD values shown in Table 2, equivalent to the RMS noise) it can be seen that the most effective way of removing the background field is to use the copper room, with the use of the shield can

improving the results by just about $1.2\%$.

We take the lowest value ($8.73\ nT$) as the resolution of the sensor. This value is already close to the digital resolution of the instrument (determined by the least-significant bit of the digital output) which, with the configuration being used, is about $3.33\ nT/LSB$.

Another standard way of measuring the capability of an instrument is by determining its noise floor. The frequency response

of the sensor's noise features a $1/f$ dependence. Since the noise density is not flat, the resolution alone does not fully describe the sensor's ability to detect low frequency signals. Thus, the noise density at $1\ Hz$ is used as the noise floor.

This is calculated by performing a Fast Fourier Transform (FFT) of the measured signal with a given DC field and calculating the value of the power spectrum at $1\ Hz$. Figure 6 shows the noise density for five different measurements taken under the same field conditions. Each measurement cycle consisted of a $30\ s$ period at a sampling frequency of $40\ Hz$ and the mean of

the five calculated values was taken, giving a noise floor of $506\ pT/\sqrt{Hz}$ @ $1\ Hz$.

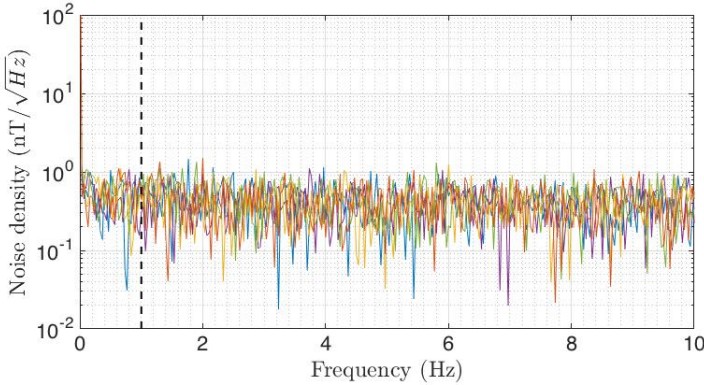

**Figure 6.** FFT of the signals measured for five different cycles over $30\ s$ with an external DC field. The level of the noise density at 1 Hz (vertical dashed line) is taken as a measure of the noise floor of the instrument.

If signals with low frequency are to be measured, over-sampling and further averaging can significantly decrease the noise level. This can be seen in Figure 7 that shows the RMS noise with respect to the number of samples used for averaging the data. Given the random nature of the data being analyzed (the sensor's noise), a simple rectangular window was applied for the averaging. For this plot data collected over a period of 100 hours (see next sub-section for a stability analysis) were used and

different number of samples were used for averaging.

Since the sampling frequency used during the data collection was 40 Hz, the plot gives the noise level that could be expected when measuring changes in the field with frequencies ranging from 40 Hz (no averaging) down to 1 Hz (averaging over 40





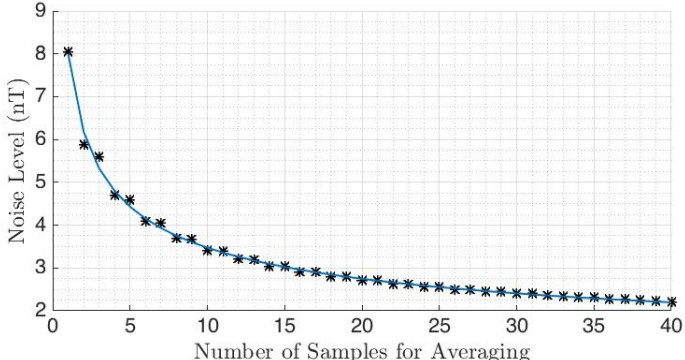

**Figure 7.** Noise level for different sizes of averaging window. The solid line represents a second order polynomial fit to the data.

samples). From the plot it can be seen that already using 4 samples for the averaging brings down the noise level to less than 5 nT, while increasing the number of samples to 20 brings the noise to about 2.7 nT and for 40 samples (which corresponds to 1 s cadence data) the noise level is 2.2 nT.

## 3.2  Stability

The stability of the sensor determines how constant the output will be under no variations of the external field (input). To measure the stability of the RM3100 magnetometer, the sensor was placed in the shield can, with no external magnetic field applied (just the residual of the Earth's field), and was set positioned so that the Z-axis of the sensor was aligned vertically with the shield can. The system was left running for 100 h. During this time, the sensor worked without any loss of data and no appreciable variation of the detected magnetic field above the noise floor was observed.

Figure 8 shows a histogram with the distribution of the measurements gathered over the $100\ h$ after removing the residual field. The fact that the distribution is normal (with a Kurtosis index of 2.86) shows the random nature of the variability observed which corresponds to the intrinsic noise of the instrument.

## 3.3  Linearity

For a sensor to produce reliable measurements, it needs to perform linearly. This means that a change in the quantity being
measured must produce a proportional change in the output of the instrument, with the proportionality being maintained throughout the whole range of the instrument.

To test the linearity of the RM3100 magnetometer, the sensor was placed inside the shield can and an external field was applied with the help of a coil aligned with the axis of interest of the instrument. The field was varied from $-100,000\ nT$ to $+100,000\ nT$ in steps of approximately $4,500\ nT$ and for each field value the sensor was exposed for 10 seconds. After this,
the median of the measured values was taken and the results obtained are shown in Figure 9.



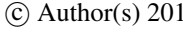

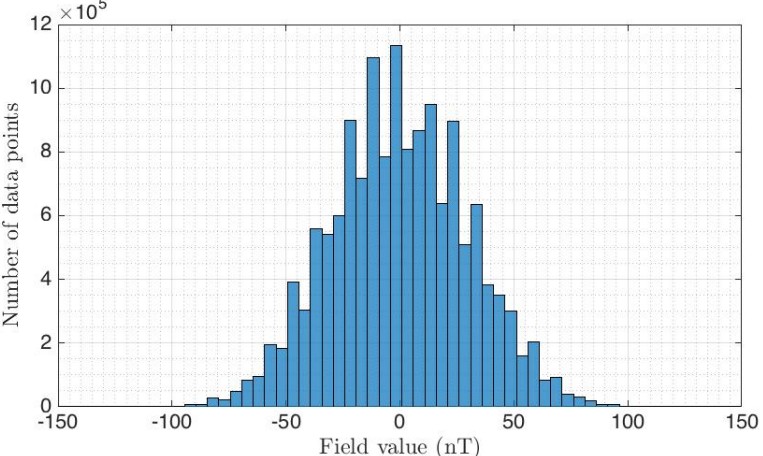

**Figure 8.** Distribution of measurements during stability analysis.

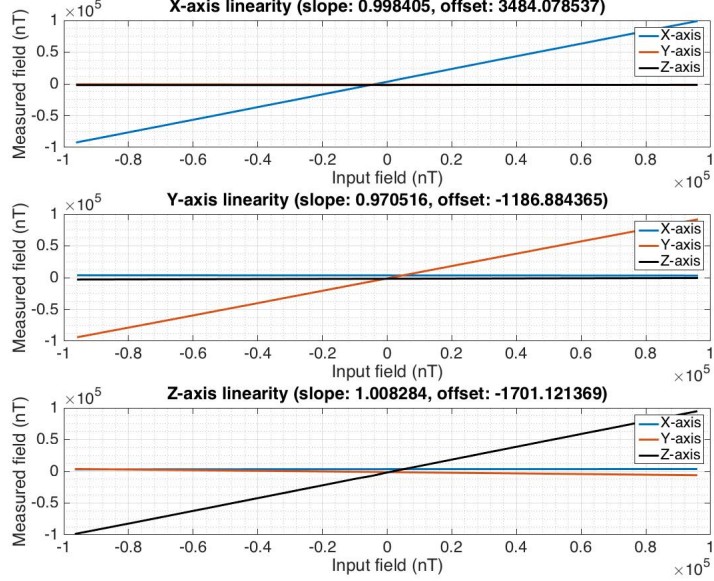

**Figure 9.** Results of the linearity test shown as the output (measured) field vs. input (applied) field for the three axes of the magnetometer.

Each panel represents a specific axis at which the external field was applied. On top of each plot, the slope and offset of a linear fit for the corresponding axis are listed. The three slopes are close to 1, varying by no more than 3%, meaning that the sensor remains linear over the whole range tested. The axis with the largest offset is the Y-axis, although the goodness of fit does not vary with respect to the other two axes. In addition, this value is larger than that reported by the manufacturer



(0.5 %), meaning that this deviation is due to a misalignment between the axis of the applied field and the coil. With the current experimental setup it is not possible to measure this misalignment and thus we take the value of 3% as an upper limit for the linearity.

It can be seen that, while the two axes not being affected should remain constant, the field in fact does change over the range

of the experiment. This is caused by a misalignment of the sensor with the axis at which the field is being applied, something that we are not correcting for in this set of tests.

## 3.4   Frequency response

For the RM3100 magnetometer to be considered for space physics applications, it must be capable of measuring magnetospheric waves. This translates into the magnetometer being able to distinctively detect signals with low amplitude and in the

ultra-low frequency (ULF) range, ideally up to 5 Hz.

In order to evaluate the frequency response of the magnetometer, the sensor was placed inside the shield can and sinusoidal signals with varying frequency between 1 and 20 $Hz$ were applied. The initial amplitude of the field for the 1 $Hz$ signal was 346 $nT$, set with a function generator with a fixed voltage. While the amplitude of the signal will change with increasing frequency due to the change in the impedance of the coil used for the generation of the field, this does not affect the results

presented here since we don't measure the RMS value of the detection, but rather the signal-to-noise ratio (SNR).

For each measurement, the noise density of the signal was calculated and two parameters were used to characterize the quality of the detection. The two parameters, dependent on the frequency of the signal, are shown in Figure 10.

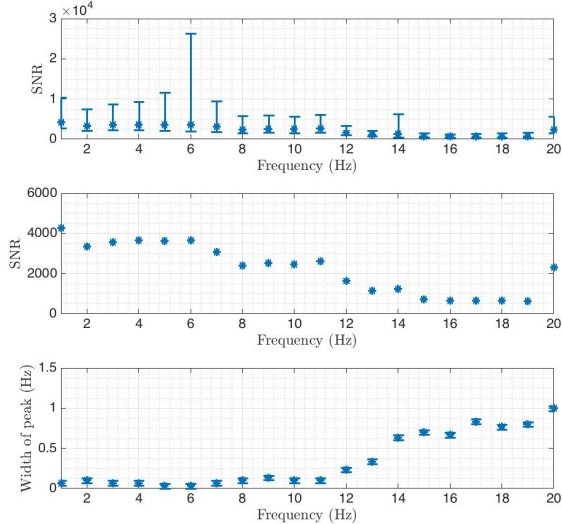

**Figure 10.** Noise level for different sizes of averaging window.





The first one is the SNR as a way to evaluate how prominent the peak of the detected signal was with respect to the background noise. The error bars are derived from the standard deviation of the signal outside the peak. The second panel shows the same plot just described but with the error bars removed to better show the trend.

The third plot shows the width of the peak in the Fourier transform. This parameter is analogous to the full width at half maximum, only that the width was calculated where the signal corresponded to 5% of the peak. The reason for this arbitrary factor is that, at half maximum, the peak is still so narrow that the corresponding frequencies cannot be resolved. This, in turn, is a consequence of the frequency bin size which, for the current experiments, is $33.3\ mHz$. This bin size corresponds to the error bars shown.

From the figure it can be seen that the response of the sensor is quite linear up to 6 Hz and has a significant decrease in both the SNR and the width of the peak at around 12 Hz. This places the performance of the sensor in a promising position to study ULF waves in the magnetosphere that can reach frequencies of about $5\ Hz$ in the case of Pc1.

Figure 11 shows four different power spectral density plots for increasing frequency of the input signal, showing the degradation of the detection, mostly visible in the increasing width of the detected peak.

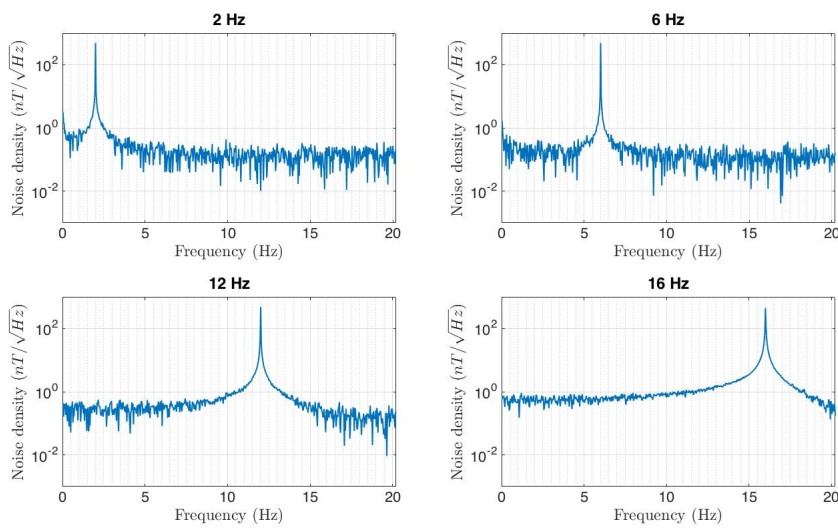

**Figure 11.** Noise density plots for four different frequencies of the input signal.

The widening of the peak at the frequency being studied arises from a distortion that appears in the measured signal as can be seen in Figure 12, where the time series of the signals corresponding to the PSD plots from Figure 11 are presented.

The distortion is an effect of the sampling frequency not being sufficiently high and it gets worse when the frequency of interest is close to the Nyquist frequency of the system (half the sampling frequency) so this effect can be avoided by increasing the sampling frequency. As mentioned before, there is a compromise between the sampling frequency and the noise floor of



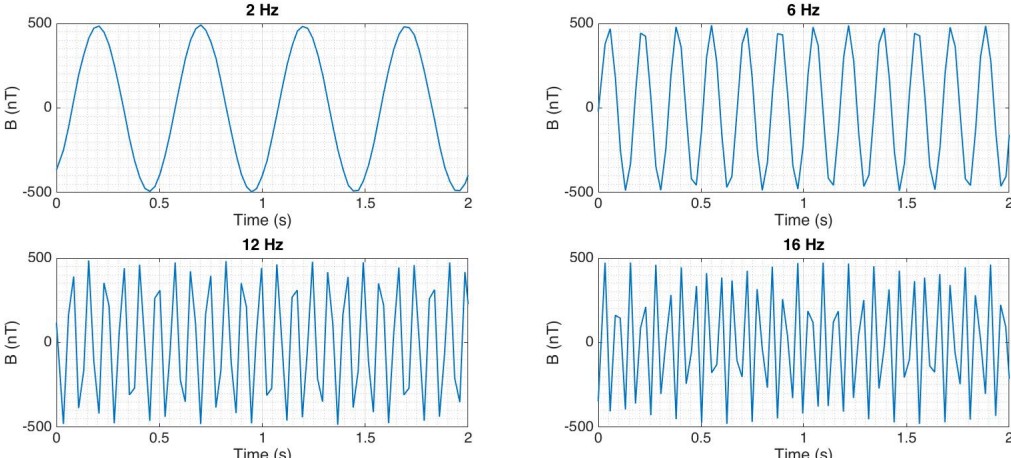

**Figure 12.** Detected signals for four different frequencies.

the instrument but the current development of a new instrument using the magneto-inductive technology at the University of Michigan will allow for higher sampling frequencies without compromising the resolution of the sensor.

## 4  Discussion

Table 3 shows a summary of the characteristics of the RM3100 magnetometer together with other magnetometers flown in
space as reference.

The list shown in Table 3 includes two top-class science magnetometers flown on major space missions (Cassini FG and MMS DFG), a recently developed miniaturized fluxgate magnetometer (Miles FG), an AMR-based magnetometer designed for small satellites (MAGIC), a COTS magnetometer (LEMI-011B) and the modified RM3100 magneto-inductive magnetometer.

One factor to take into account when comparing different instruments is the time when instruments were built. Technology
advances that happen in less than a decade significantly change the ability to either improve an instrument's overall performance or decrease its size and mass while maintaining a similar performance.

In that sense, the sensors listed in Table 3 cover an extended period of time. Cassini was launched in 1997 and performed its orbit insertion maneuver in 2004. This places the construction of its fluxgate magnetometer (Dougherty et al., 2004) more than two decades ago. The Magnetospheric Multiscale (MMS) mission (Russell et al., 2016) was launched in 2015 and includes
several magnetometers. The characteristics listed in the table correspond to the digital fluxgate. Apart from the differences in design, from Table 3 it can be seen that these two decades allowed for a significant reduction in mass (a factor of 3) and power consumption (more than a factor of 10) while achieving similar noise performances.

The third sensor in the table (Miles et al., 2016) was developed with the idea of providing a scientific fluxgate small and light enough to be placed on a CubeSat. The reduction in mass, in comparison with the MMS DFG is not significant, but they also





**Table 3.** Performance comparison of different magnetometers.

| Sensor | Range [$nT$] | Freq. [$Hz$] | Noise [$nT/\sqrt{Hz}$] | Mass [$g$] | Dimensions [$mm$] | Power [$W$] | Cost ($) |
|---|---|---|---|---|---|---|---|
| MMS DFG[a] | ±10,500 | 128 | < 0.008 @1 $Hz$ | 140 | 42.4 x 44.3 x 48.7 (s) | 450 | > 1 M |
| | | | | | 700 x 1100 (e) | | |
| Cassini FG[b] | ±44,000 | 32 | < 0.005 @1 $Hz$ | 440* | - | 7.5 | > 1 M |
| Miles FG[c] | ±65,536 | 40 | 0.2 @1 $Hz$ | 112 | 36 x 32 x 28 (s) | 0.4 | - |
| | | | | | 96 x 91 (e) | | |
| MAGIC[d] | ±57,500 | 20 | 0.15 @1 $Hz$ | 104 | 20 x 20 x 5 (s) | 0.5 | > 1 k |
| | | | | | 90 x 96 (e) | | |
| LEMI-011B[e] | ±60,000 | 20 | 0.7 @12.83 $Hz$ | ≤ 120 | 50 x 16 x 16 (s) | 0.03 | ∼ 600 |
| | | | | | 55 x 50 (e) | | |
| MOURA[f] | ±65,000 | - | 0.85 @0.5 $Hz$ | 72 | 150 x 30 x 15 (s) | 0.4 | ∼ 500 |
| RM3100 | ±100,000 | 40 | 0.5 @1 $Hz$ | < 3 | 25.4 x 25.4 x 9.6 (s+e) | < 0.01 | ∼ 30 |

[a](Russell et al., 2016), [b](Dougherty et al., 2004), [c](Miles et al., 2016), [d](Brown et al., 2014),

[e](Matandirotya et al., 2013), [f](Díaz-Michelena et al., 2015), . * Mass only corresponds to the sensor, no electronics.

(s) stands for sensor and (e) stands for electronics board.

designed a low-mass boom. Similarly, the reduction in power consumption is still moderate and the noise floor is significantly higher than that of the Cassini FG or the MMS DFG. However, the fact that a fluxgate magnetometer was specifically designed with the idea of CubeSat missions in mind is a reflection of the attention that small satellite missions are gaining and is an important step towards the further miniaturization and performance improvement of scientific instruments.

5      The MAGIC sensor (Brown et al., 2014) was built for the TRIO-CINEMA space weather mission, formed of three 3U CubeSats, and has already returned useful scientific data (Archer et al., 2015). The instrument's noise floor is slightly better than that of the Miles FG and presents a small improvement in mass but with a small increase in power consumption. The fact that the mission has already provided in-flight results represents an important advance in the development of science missions based on small satellites.

10     The LEMI-011B is a commercial fluxgate magnetometer produced by Lviv Institute of Space Research. Although not specifically designed for space science applications, as assessed by Matandirotya et al. (2013), the measurement capabilities, together with the low mass and power consumption, make this a suitable candidate to study some of the ULF waves with largest amplitude (Pc3 to Pc5).

The MOURA sensor (Díaz-Michelena et al., 2015) was developed to measure the surface magnetic field at Mars as a part 15     of the Mars MetNet Precursor Mission. The instrument's requirements were derived from the specific environment to be encountered at Mars and it makes use of shielded COTS components.




When comparing the different parameters with the RM3100 magnetometer, it can be seen that the main strength of the latter so far is its size, mass, power consumption and cost (about \$30 for the COTS PNI compared to about \$600 for the LEMI-011B and several hundreds of thousands for large-scale mission magnetometers).

Performance measurements such as the noise floor and the resolution can still be improved but already are at a level that would allow the sensor to measure waves in the Pc1 to Pc5 range and represent an improvement with respect to other COTS sensors. In general, this study allowed us to identify weak spots in the design of the RM3100 (the instrument was designed with compass applications in mind) that limit its performance and we are currently working on the development of a new magneto-inductive magnetometer addressing those limitations.

## 5    Conclusions and current developments

Based on the results presented in this paper and the measurement requirements inposed by the nature of ULF waves, the magneto-inductive technology arises as a promising measurement principle for space- and ground-based platforms to study DC magnetic fields and magnetospheric waves. Due to the relatively large range of frequencies, the magneto-inductive magnetometer can cover the Pc1 range, something that is often not possible with other types of sensors, needing a separate search coil magnetometer instead.

The use of COTS components potentially enables future missions throughout the solar system to employ large constellations of CubeSats allowing for rapid multipoint sampling. This approach would greatly aid our understanding of the large-scale dynamics of planetary magnetospheres and how different processes affect different regions in space.

The magneto-inductive technology, with its small size and weight as well as low power cosumption and costs, is a promising candidate for these types of missions. For the same reasons, it could be used to provide an extensive network of autonomous ground-based magnetometers to complement in situ observations. By packaging the sensor together with a power source such as solar panels or a set of batteries, data could be collected over extended periods of time at very low costs. Additionally, it can be operated in extreme polar environments where long winters constrain operations.

One of the current limitations of the sensor as presented here is the resolution which, without applying oversampling techniques is about 8 nT. While these resolution levels are already sufficient to perform studies of large-scale currents and magnetospheric waves in the Pc3 to Pc5 range, in general, sub-nT sensitivities would be desired to better measure the small changes in the magnetic field produced by changes in solar wind conditions.

The noise floor lies at $506 \, pT/\sqrt{Hz} \, @ \, 1 \, Hz$, providing a better performance compared to other COTS sensors (Matandirotya et al., 2013) and comparable to some sensors specifically developed for small satellite missions (e.g., Miles et al., 2016; Brown et al., 2014).

In terms of performance, there are a variety of ways to improve the RM3100's capabilities. Currently, development efforts are being carried out at the University of Michigan in order to improve the resolution and noise floor of the sensor by changing different parameters of the basic design such as the clock frequency and its dynamic range. In addition, studies aimed at using



different coils are being undertaken as well as the inclusion of several magnetometers in a single board in order to reduce the noise by over-sampling the signals.

With these changes, we expect the newly developed magnetometer to have sub-nT resolution while maintaining or even reducing the size and mass of the RM3100 used for the tests presented in this paper.

5 *Competing interests.* The authors declare that they have no conflict of interest.

*Acknowledgements.* This work was supported by a NASA Heliophysics Technology and Instrument Development for Science grant (NNX16AH47G) and a NASA Small Spacecraft Technology Program grant (NNX16AT35A).





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
