# Peer review of "Investigation of a low-cost magneto-inductive magnetometer for space science applications"

_Geoscientific Instrumentation, Methods and Data Systems, 2017_

## Referee Comment (RC1) · DMM Miles (Referee) · 22 Dec 2017

General Comments:

This paper investigates the suitability of a commercial magneto-inductive sensor for use in geo and space physics applications particularly on small and low-cost platforms such as CubeSats. The tests results presented suggest that the sensor compares well to other types of magnetometers in the literature in terms of magnetic range, frequency range, noise, mass, power, volume and cost. The paper aligns well with the scope of GI and is timely given recent trend towards small and low-cost platforms. The results of the paper are, in general, sound and backed by the data. However, to more fully evaluate

the suitability of the sensor for geophysical applications some additional details and analysis would be appropriate.

In Section 2, the description of how the external magnetic field manifests in the output of the oscillator is sparse and would benefit from being shown explicitly by, for example, oscilloscope captures under different applied fields. The authors note that, in this manuscript, their optimization of the sensor was software only. Could the authors suggest avenues to further improve the sensor performance via hardware modification – for example by increasing the oscillator frequency or the base clock against which the output of the sensor is compared?

In Section 3.3, the description of "Linearity" matches what I would describe as the Gain Error/Sensitivity of the sensor (i.e., nT_measured / nT_applied). This is a useful metric to calibrate the sensor but does capture the deviation of the instruments performance from linear. To do this, the authors should fit and subtract a linear trend to the results of each axis and plot the residual to show any non-linear trend. Quantitatively, the authors should also present the RMS deviation of the measured data from the fit linear trend.

Could the authors estimate the offset/zeros of the sensor using a simple technique such as 180 degree sensor rotations? The offsets shown in Figure 9 suggest that the zeros could be large, on the order of 1000 nT, but they would need to be separated from the residual external magnetic field in the test chamber.

Specific Comments:

Page 2, Lines 1-4. Reference specific current and planned missions

Page 2, Table 1. Add approximate amplitude range for each wave category to motivate the required magnetometer performance.

Page 5, Line 1. Add a reference describing the magneto-inductive principle.

Page 5, Figure 1. Note what the red boxes represent in the figure caption and label the X, Y, and Z axes.
Page 6, Figure 2. Expand the figure to show explicitly how the output of the oscillator is measured and how that corresponds to the H_E

Page 6, Figure 3. Provide a more detailed description of what is being presented and define all terms used in the figure.

Page 9, Figure 6. The power spectrum appears essentially flat with frequency rather than exhibiting the 1/f dependence referenced in the text. Recreating the plot from a longer period of data, applying some averaging, and using log-log axes would make it easier to understand shape of the noise density. Also, if the sensor is sampling at 40 Hz why does the frequency range stop at 10 Hz rather than the 20 Hz Nyquist?

Page 10, Section 3.2. It would be interesting to see a raw and low-pass filtered time series of the 100 hour data set to see if there are any significant trends.

Page 11, Figure 8. The distribution shows some large discrepancies in adjacent bins which is somewhat surprising given the large number of data points (∼10ˆ5). Could the authors comment on the potential cause? Is this statistical or suggestive of a digitization artifact similar to missing codes.

Page 12, Section 3.4. It would be interesting to see the signal amplitude as well as the calculated SNR.

Page 12/13, Figures 10/11. The large error bars in the SNR calculation might be reduced by the application of an appropriate window function to better separate the test signal from the background. e.g., Heinzel et al., 2002.

Page 12, Figure 10. Is this the correct caption for the figure?

Page 16, Lines 12-14. The difference in sensitivity at Pc1 frequencies between the presented instrument and a typical search coil magnetometer is not insignificant – e.g., Figure 5 in Hospodarsky, (2016)

Technical Corrections:

Page 5, Line 11. Define RL

Page 5, Line 11, Define A/D

Page 5, Line 15. Define 'H'

Page 6, Line 3. Here the sampling rate is 'approximately 40 Hz' elsewhere it is just referred to as '40 Hz'. Please clarify.

Page 8, Table 2. Add units to the 'STD' column.

Page 8, Figure 5. Add the Configuration # from Table 2 to the legend.

Page 15, Table 3. It seems unlikely that the MMS DFG consumes 450 W.

Page 15, Table 3. The miniature fluxgate from Miles et al. 2016 produces 100 Hz data.

References:

Heinzel, G., Rüdiger, A., & Schilling, R. (2002). Spectrum and spectral density estimation by the Discrete Fourier transform (DFT), including a comprehensive list of window functions and some new at-top windows.

Hospodarsky, G. B. (2016). Spaced-‐based search coil magnetometers. Journal of Geophysical Research: Space Physics, 121(12).

---

## Author Comment (AC1) · 5 Jan 2018

First of all, thank you very much for reading the paper and providing very useful comments. We went through all the comments and addressed them, so please find below our replies for the comments that were not addressed specifically as suggested:

1. We are not including outputs of the oscillator as suggested, since the field is calculated based on the difference between the charge and discharge times of the RL circuit shown in the paper. The oscillator, in the commercial version, is part of the ASIC controller so the output cannot be measured.

[Figure]

In terms of hardware improvement, as discussed in the future work section, we are currently working on a new implementation. Two components are key to improve the performance of the sensor: the coils and the clock (specifically the frequency). We cannot provide details of performance yet since the new instrument is in the early stages of development, but based on initial calculations, we expect the improvement in resolution to scale linearly with the clock frequency.

2. Since we haven't performed any test on the alignment of the coils, we don't aim at giving a full characterization of the offset of the coils. The suggestion of performing a set of tests with rotation of 180 degrees is a very good one, but the test setup with which the measurements were taken does not allow for a precise rotation along a single axis. When the unit developed at the University of Michigan is ready, we expect to be able to perform these more thorough tests by using external facilities.

3. Figure 6 was not expanded because, as explained in comment 1, the way the fields are measured involved the measurement of the output of a timer. This is a very simplistic schematic of the electronics and we feel that a more detailed schematic would not help convey the working principle, which is better understood through a text description.

4. The amplitude of the signal is not included in the paper because at the current stage we are not able to distinguish between the attenuation caused by the sensor itself and that caused by the change in the impedance of the coil used to generate the field with frequency. I am attaching the plot for reference, but we feel that including it in the paper would lead to confusion, hence we decided to just include the SNR.

5. The error bars in Figure 10 are a consequence of the variability of the random noise of the instrument. Instead of presenting them as a function of the standard deviation of the signal, we used the standard error, which reduces the error bars significantly.

Note: a significant change with respect to the previous version is the noise floor. We were using a simple Fourier transform of the output instead of using the Fourier

transform of the autocorrelation function, as defined in Heinzel et al. (2012). This brought the noise floor down to 4 pT/sqrt(Hz) which is extremely good. It seems like different authors treat noise floor in different ways, so any comments on this respect are more than welcome.

Please also note the supplement to this comment:
https://www.geosci-instrum-method-data-syst-discuss.net/gi-2017-53/gi-2017-53-AC1-supplement.pdf
* * *
[Figure]

**Fig. 1.**

**Supplement:**

**Investigation of a low-cost magneto-inductive magnetometer for space science applications**

Leonardo H. Regoli[1], Mark B. Moldwin[1], Matthew Pellioni[1], Bret Bronner[2], Kelsey Hite[1], Arie Sheinker[3], and Brandon M. Ponder[4]

[1]Climate and Space Sciences and Engineering, College of Engineering, University of Michigan, Ann Arbor, USA
[2]Space Physics Research Laboratory, College of Engineering, University of Michigan, Ann Arbor, USA
[3]Magnetic Sensing, Soreq Nuclear Research Center, Israel
[4]Nissan Technical Center North America (NTCNA), USA

*Correspondence to:* Leonardo H. Regoli (lregoli@umich.edu)

**Abstract.** A new sensor for measuring low-amplitude magnetic fields that is ideal for small spacecraft is presented. The novel measurement principle enables the fabrication of a low-cost sensor with low power consumption and with measuring capabilities that are comparable to recent developments for CubeSat applications. The current magnetometer, a software-modified version of a commercial sensor, is capable of detecting fields with amplitudes as low as $8.7\ nT$ at $40\ Hz$ and $2.7\ nT$ at $1\ Hz$, with a noise floor of $4\ pT/\sqrt{Hz}$ @ $1\ Hz$. The sensor has a linear response to less than $3\%$ over a range of $\pm 100,000\ nT$. All of these features make the magneto-inductive principle a promising technology for the development of magnetic sensors for both space-borne and ground-based applications to study geomagnetic activity.

**1 Introduction**

Magnetic fields are a ubiquitous feature of our solar system and of key importance for geophysical, magnetospheric and heliospheric investigations. The sun produces the interplanetary magnetic field (IMF) and many of the planets and moons throughout the solar system produce their own magnetic fields through dynamo and magneto-inductive response processes. Even where no internally produced magnetic field is present, for example, Mars, or Venus, the IMF plays a major role in how planets and smaller bodies interact with the solar wind.

At Earth, the measured field is a combination of the internal dynamo-generated field and perturbations that occur in space, particularly during substorm and geomagnetic storm processes. These processes are governed by the direction of the IMF and the dynamic pressure exerted by the solar wind at any given time (e.g., Moldwin, 2008). The enhancement of the particle fluxes in the ring current during a geomagnetic storm causes the measured magnetic field strength at the surface of the Earth to decrease. This is quantified by the so-called disturbance storm time (Dst) index, which is determined by a network of low-latitude magnetometers (e.g., Hamilton et al., 1988; Liemohn et al., 2001).

The dynamic nature of planetary magnetospheres makes it extremely difficult, if not impossible, to understand their structure without the help of a magnetometer with sufficient resolution, dynamic range, and bandwidth, to discriminate between the different regions inside the magnetosphere and identify the magnetic signature of plasma flows that are governed by global and

local circulation patterns. For this reason, magnetometers have been a key tool in magnetospheric investigations throughout the history of their study and continue to be indispensable. Critically, current (e.g. MMS, Russell et al. (2016) and Cluster, Balogh et al. (1997)) and planned (e.g. MagCon, Kepko and Le (2004)) investigations of multi-scale dynamic features throughout the solar system continue to drive the need for greater numbers of magnetometers with state of the art capabilities.

**1.1 ULF waves in the magnetosphere**

The Earth's magnetosphere, whose field strength varies from about $60,000\ nT$ in polar LEO orbit to about $100\ nT$ at geosynchronous orbit, have different wave populations present with frequencies ranging from a few mHz to a few Hz on both the day- and nightside. Traditionally, the continuous pulsations which are denoted by Pc1-5 can be divided into categories that are characterized by a given frequency range as summarized in Table 1 (e.g., Jacobs et al., 1964; Fraser, 2007; Menk, 2011).

**Table 1.** ULF waves in the magnetosphere.

| Wave | Frequency | Amplitude |
|------|-----------|-----------|
| Pc1 | $0.2 - 5\ Hz$ | $\sim 0.1\ nT$ |
| Pc2 | $0.1 - 0.2\ Hz$ | $\sim 0.5\ nT$ |
| Pc3 | $22 - 100\ mHz$ | $\sim 0.5\ nT$ |
| Pc4 | $7 - 22\ mHz$ | $\sim 10\ nT$ |
| Pc5 | $1 - 7\ mHz$ | $\sim 50\ nT$ |

[revised manuscript text omitted]
 3, where $\mu(H)$ represents the induction curve of the sensing coils, $\tau_P$ and $\tau_N$ represent the positive and negative bias charge-discharge time respectively, $H_L$ and $H_H$ the low and high charge threshold, $H_S$ the is the positive-to-negative bias (adjusted to take advantage of the symmetry of the induction curve) and $H_E$ is the inductance bias caused by the external field.

[Figure]

**Figure 2.** Schematics of the electronics involved in the magneto-inductive technology (from Leuzinger and Taylor (2010)).

When no external field is applied, the charge and discharge times calculated at both polarities are the same ($\tau_P = \tau_N$). However, if an external field is applied ($H_E$), the working region along the curve will be shifted in one direction, and consequently the charge and discharge times will no longer be equal ($\tau_P > \tau_N$ for the specific case shown in Figure 3). This time difference is proportional to the applied external field.

[Figure]

**Figure 3.** The induction in the coils as a function of applied magnetic field (top) and the traces of the oscillating current in the solenoid and the period for positive and negative bias polarity (bottom) (from Leuzinger and Taylor (2010)).

5    An applied magnetic field causes a constant offset in the coils' field strength, the polarity of which is determined by the direction of the field. This offset causes the average permeability and therefore inductance to be lower in one direction and larger in the other yielding a corresponding difference in the time required to complete the minor B-H loops in each direction.

By integrating over many such minor loops in each direction, the time difference, and therefore available resolution, can be enhanced to any desired level subject to integrated noise sources.

In the commercial version of the sensor, the number of loops used for integration is controlled by a register called cycle count and, for all the experiments presented in this paper, this value was set to 800. The value of this register is inversely proportional to the sampling frequency which, for this paper was of approximately $40\ Hz$ (although set to sample at $40\ Hz$, the real sampling frequency is slightly lower than this by less than $1\ 
[revised manuscript text omitted]

    Figure 10 shows the non-linearity for each individual axis, taken as the difference between the corresponding measurements

20    taken by the sensor and the linear fit to the data shown in Figure 9 for each axis. All the axes show a similar trend, with the maximum deviation from linearity happening at larger fields and a sinusoidal-like pattern at the center of the measurement

[Figure]

**Figure 9.** Results of the linearity test shown as the output (measured) field vs. input (applied) field for the three axes of the magnetometer.

range. In this central region, the maximum deviation is of $200 \ nT$ at a field strength of $40,000 \ nT$, which represents an error of $0.5 \ \%$.

[Figure]

**Figure 10.** Non-linearity of the sensor for each axis defined as the difference betweeen the mesured field and a linear fit to the data.

**3.4 Frequency response**

For the RM3100 magnetometer to be considered for space physics applications, it must be capable of measuring magnetospheric waves. This translates into the magnetometer being able to distinctively detect signals with low amplitude and in the ultra-low frequency (ULF) range, ideally up to 5 Hz.

5    In order to evaluate the frequency response of the magnetometer, the sensor was placed inside the shield can and sinusoidal signals with varying frequency between 1 and 20 $Hz$ were applied. The initial amplitude of the field for the 1 $Hz$ signal was 346 $nT$, set with a function generator with a fixed voltage. While the amplitude of the signal will change with increasing frequency due to the change in the impedance of the coil used for the generation of the field, this does not affect the results presented here since we don't measure the RMS value of the detection, but rather the signal-to-noise ratio (SNR).

10    For each measurement, the noise density of the signal was calculated and two parameters were used to characterize the quality of the detection. The two parameters, dependent on the frequency of the signal, are shown in Figure 11.

[Figure]

**Figure 11.** Signal-to-noise ratio (top panel) and width of the peak (bottom panel) detected by the RM3100 as a function of the input signal frequency.

[revised manuscript text omitted]

---

## Referee Comment (RC2) · P. Mahavarkar (Referee) · 5 Feb 2018

General comments:

The authors have done a very good work and I believe the manuscript deserves publication in the journal provided some minor changes are introduced.

Detailed comments:

Page No. 2, Line No. 8

"The Earth's magnetosphere, whose field strength varies from about 60, 000 nT in polar LEO orbit to about 100 nT at geosynchronous orbit, have different wave populations

present with frequencies ranging from a few mHz to a few Hz on both the day- and nightside."

It would be better if the authors mention the approximate time duration for example day (______) and night (____).

Page No. 2, Line No. 24 and 26

"The use of ground-based magnetometers, depending on their distribution around the globe and in combination with global models of the magnetosphere, can also shed light on how global the distrubances are, by correlating the signals observed at different latitudes with the length of the corresponding magnetic field lines."

What do the authors mean by the length of the corresponding magnetic field lines? Secondly please correct the spelling 'distrubances'.

Page No. 3, Line No. 20

"sensors have predominantly been used for space missions, namely fluxgate and helium magnetometers. However, due to their high fabrication costs, relatively large size and high power needs,"

Please tabulate the power consumption for the fluxgate and helium magnetometers and the approximate weight.

Page No. 3, Line No. 22

"One approach is to miniaturize fluxgate magnetometers,"

Miniaturize in terms of weight or size? Please mention the same.

Page No. 4, Line No. 23

"(SWAP+C)"

SWAP+C ?

Page No. 5, Line No. 3 and 4

"The modifications thus far consist of optimization of internal parameters of the sensor that allowed us to improve its performance."

In my opinion sentence is not correct.

Page No. 5, Line No. 6

"The COTS version shown in Figure 1 consists of the orthogonal coils (indicated with red rectangles),"

Please insert a better quality image where the coils are visible. It is difficult to visualize the components.

Page No. 5, Line No. 8

"CommBoard introduces a constant interference of up to several hundreds of nT"

Please mention the approximate range.

Page No. 5, Line No. 17

"where k is a property of the coil,"

What property?

Page No. 12, Line No. 15

"presented here since we don't measure"

Correction: 'do not'

Page No. 16, Line No. 13

"needing a separate search"

The word 'needing' has to be replaced by an appropriate word

---

## Author Comment (AC2) · 5 Feb 2018

First of all, we want to thank the reviewer for taking the time to read the paper and provide useful feedback. Please find below our replies to the comments:

- Added reference to papers related to occurrence of ULF waves in day- and nightside. The specific characteristics of different ULF waves is a very broad subject of study. Since our paper only presents an instrument capable of measuring those waves, we feel that referring the reader to the relevant literature is the appropriate way to keep the paper focused.

[Figure]

- We refer to the length of the magnetic field lines because, under a closed field line configuration, the frequency of the waves is correlated to this parameter.

- The power consumption and weight of a couple of fluxgate magnetometers are listed in Table 3. We did not include comparison with any helium magnetometer, but these parameters are similar to those of a fluxgate.

- Added specific mention to size and weight when referring to miniaturization of fluxgate magnetometers.

- SWAP+C refers to reduced size, weight, power and costs, as explained in the text before the acronym.

- Changed "internal parameters" to "software parameters" to make it clear that we mean software, and not hardware, modifications.

- The coils are difficult to see because their color is similar to that of the board. I tried increasing the brightness of the image and decreasing the contrast a bit. Please let me know if that is better.

- Measured value of interference introduced by CommBoard now indicated.

- "k" is the conversion factor of the coil (how much field is generated given a current flow). This has been added to the text for clarification.

Please also note the supplement to this comment:
https://www.geosci-instrum-method-data-syst-discuss.net/gi-2017-53/gi-2017-53-AC2-supplement.pdf

———————————————

[Figure]

**Supplement:**

**Investigation of a low-cost magneto-inductive magnetometer for space science applications**

Leonardo H. Regoli[1], Mark B. Moldwin[1], Matthew Pellioni[1], Bret Bronner[2], Kelsey Hite[1], Arie Sheinker[3], and Brandon M. Ponder[4]

[1]Climate and Space Sciences and Engineering, College of Engineering, University of Michigan, Ann Arbor, USA
[2]Space Physics Research Laboratory, College of Engineering, University of Michigan, Ann Arbor, USA
[3]Magnetic Sensing, Soreq Nuclear Research Center, Israel
[4]Nissan Technical Center North America (NTCNA), USA

*Correspondence to:* Leonardo H. Regoli (lregoli@umich.edu)

**Abstract.** A new sensor for measuring low-amplitude magnetic fields that is ideal for small spacecraft is presented. The novel measurement principle enables the fabrication of a low-cost sensor with low power consumption and with measuring capabilities that are comparable to recent developments for CubeSat applications. The current magnetometer, a software-modified version of a commercial sensor, is capable of detecting fields with amplitudes as low as $8.7\ nT$ at $40\ Hz$ and $2.7\ nT$ at $1\ Hz$, with a noise floor of $4\ pT/\sqrt{Hz}$ @ $1\ Hz$. The sensor has a linear response to less than $3\%$ over a range of $\pm 100,000\ nT$. All of these features make the magneto-inductive principle a promising technology for the development of magnetic sensors for both space-borne and ground-based applications to study geomagnetic activity.

**1 Introduction**

Magnetic fields are a ubiquitous feature of our solar system and of key importance for geophysical, magnetospheric and heliospheric investigations. The sun produces the interplanetary magnetic field (IMF) and many of the planets and moons throughout the solar system produce their own magnetic fields through dynamo and magneto-inductive response processes. Even where no internally produced magnetic field is present, for example, Mars, or Venus, the IMF plays a major role in how planets and smaller bodies interact with the solar wind.

At Earth, the measured field is a combination of the internal dynamo-generated field and perturbations that occur in space, particularly during substorm and geomagnetic storm processes. These processes are governed by the direction of the IMF and the dynamic pressure exerted by the solar wind at any given time (e.g., Moldwin, 2008). The enhancement of the particle fluxes in the ring current during a geomagnetic storm causes the measured magnetic field strength at the surface of the Earth to decrease. This is quantified by the so-called disturbance storm time (Dst) index, which is determined by a network of low-latitude magnetometers (e.g., Hamilton et al., 1988; Liemohn et al., 2001).

The dynamic nature of planetary magnetospheres makes it extremely difficult, if not impossible, to understand their structure without the help of a magnetometer with sufficient resolution, dynamic range, and bandwidth, to discriminate between the different regions inside the magnetosphere and identify the magnetic signature of plasma flows that are governed by global and

local circulation patterns. For this reason, magnetometers have been a key tool in magnetospheric investigations throughout the history of their study and continue to be indispensable. Critically, current (e.g. MMS, Russell et al. (2016) and Cluster, Balogh et al. (1997)) and planned (e.g. MagCon, Kepko and Le (2004)) investigations of multi-scale dynamic features throughout the solar system continue to drive the need for greater numbers of magnetometers with state of the art capabilities.

**1.1 ULF waves in the magnetosphere**

The Earth's magnetosphere, whose field strength varies from about $60,000\ nT$ in polar LEO orbit to about $100\ nT$ at geosynchronous orbit, have different wave populations present with frequencies ranging from a few mHz to a few Hz on both the day- and nightside (e.g., Sakurai et al., 1999; Rae and Watt, 2016). Traditionally, the continuous pulsations which are denoted by Pc1-5 can be divided into categories that are characterized by a given frequency range as summarized in Table 1 (e.g., Jacobs et al., 1964; Fraser, 2007; Menk, 2011).

**Table 1.** ULF waves in the magnetosphere.

| Wave | Frequency | Amplitude |
|------|-----------|-----------|
| Pc1 | $0.2 - 5\ Hz$ | $\sim 0.1\ nT$ |
| Pc2 | $0.1 - 0.2\ Hz$ | $\sim 0.5\ nT$ |
| Pc3 | $22 - 100\ mHz$ | $\sim 0.5\ nT$ |
| Pc4 | $7 - 22\ mHz$ | $\sim 10\ nT$ |
| Pc5 | $1 - 7\ mHz$ | $\sim 50\ nT$ |

[revised manuscript text omitted]
 3, where $\mu(H)$ represents the induction curve of the sensing coils, $\tau_P$ and $\tau_N$ represent the positive and negative bias charge-discharge time respectively, $H_L$ and $H_H$ the low and high charge threshold, $H_S$ the is the positive-to-negative bias (adjusted to take advantage of the symmetry of the induction curve) and $H_E$ is the inductance bias caused by the external field.

[Figure]

**Figure 2.** Schematics of the electronics involved in the magneto-inductive technology (from Leuzinger and Taylor (2010)).

When no external field is applied, the charge and discharge times calculated at both polarities are the same ($\tau_P = \tau_N$). However, if an external field is applied ($H_E$), the working region along the curve will be shifted in one direction, and consequently the charge and discharge times will no longer be equal ($\tau_P > \tau_N$ for the specific case shown in Figure 3). This time difference is proportional to the applied external field.

[Figure]

**Figure 3.** The induction in the coils as a function of applied magnetic field (top) and the traces of the oscillating current in the solenoid and the period for positive and negative bias polarity (bottom) (from Leuzinger and Taylor (2010)).

5    An applied magnetic field causes a constant offset in the coils' field strength, the polarity of which is determined by the direction of the field. This offset causes the average permeability and therefore inductance to be lower in one direction and larger in the other yielding a corresponding difference in the time required to complete the minor B-H loops in each direction.

By integrating over many such minor loops in each direction, the time difference, and therefore available resolution, can be enhanced to any desired level subject to integrated noise sources.

In the commercial version of the sensor, the number of loops used for integration is controlled by a register called cycle count and, for all the experiments presented in this paper, this value was set to 800. The value of this register is inversely proportional to the sampling frequency which, for this paper was of approximately $40\ Hz$ (although set to sample at $40\ Hz$, the real sampling frequency is slightly lower than this by less than $1\ 
[revised manuscript text omitted]

     Figure 10 shows the non-linearity for each individual axis, taken as the difference between the corresponding measurements

20  taken by the sensor and the linear fit to the data shown in Figure 9 for each axis. All the axes show a similar trend, with the maximum deviation from linearity happening at larger fields and a sinusoidal-like pattern at the center of the measurement

[Figure]

**Figure 9.** Results of the linearity test shown as the output (measured) field vs. input (applied) field for the three axes of the magnetometer.

range. In this central region, the maximum deviation is of $200\ nT$ at a field strength of $40,000\ nT$, which represents an error of $0.5\ \%$.

[Figure]

**Figure 10.** Non-linearity of the sensor for each axis defined as the difference betweeen the mesured field and a linear fit to the data.

**3.4 Frequency response**

For the RM3100 magnetometer to be considered for space physics applications, it must be capable of measuring magneto-spheric waves. This translates into the magnetometer being able to distinctively detect signals with low amplitude and in the ultra-low frequency (ULF) range, ideally up to 5 Hz.

In order to evaluate the frequency response of the magnetometer, the sensor was placed inside the shield can and sinusoidal signals with varying frequency between 1 and 20 $Hz$ were applied. The initial amplitude of the field for the 1 $Hz$ signal was 346 $nT$, set with a function generator with a fixed voltage. While the amplitude of the signal will change with increasing frequency due to the change in the impedance of the coil used for the generation of the field, this does not affect the results presented here since we do not measure the RMS value of the detection, but rather the signal-to-noise ratio (SNR).

For each measurement, the noise density of the signal was calculated and two parameters were used to characterize the quality of the detection. The two parameters, dependent on the frequency of the signal, are shown in Figure 11.

[Figure]

**Figure 11.** Signal-to-noise ratio (top panel) and width of the peak (bottom panel) detected by the RM3100 as a function of the input signal frequency.

[revised manuscript text omitted]

5    The use of COTS components potentially enables future missions throughout the solar system to employ large constellations of CubeSats allowing for rapid multipoint sampling. This approach would greatly aid our understanding of the large-scale dynamics of planetary magnetospheres and how different processes affect different regions in space.

The magneto-inductive technology, with its small size and weight as well as low power cosumption and costs, is a promising candidate for these types of missions. For the same reasons, it could be used to provide an extensive network of autonomous
10   ground-based magnetometers to complement in situ observations. By packaging the sensor together with a power source such as solar panels or a set of batteries, data could be collected over extended periods of time at very low costs. Additionally, it can be operated in extreme polar environments where long winters constrain operations.

One of the current limitations of the sensor as presented here is the resolution which, without applying oversampling techniques is about 8 nT. While these resolution levels are already sufficient to perform studies of large-scale currents and magne-
15   tospheric waves in the Pc3 to Pc5 range, in general, sub-nT sensitivities would be desired to better measure the small changes in the magnetic field produced by changes in solar wind conditions.

The noise floor lies at $4 \ pT/\sqrt{Hz}$ @ $1 \ Hz$, providing excellent performance compared to other sensors, including both commercial (Matandirotya et al., 2013) and specifically developed for small satellite missions (e.g., Miles et al., 2016; Brown et al., 2014).

20   In terms of performance, there are a variety of ways to improve the RM3100's capabilities. Currently, development efforts are being carried out at the University of Michigan in order to improve the resolution and noise floor of the sensor by changing different parameters of the basic design such as the clock frequency and its dynamic range. In addition, studies aimed at using different coils are being undertaken as well as the inclusion of several magnetometers in a single board in order to reduce the noise by over-sampling the signals.

25   With these changes, we expect the newly developed magnetometer to have sub-nT resolution while maintaining or even reducing the size and mass of the RM3100 used for the tests presented in this paper.

*Competing interests.* The authors declare that they have no conflict of interest.

*Acknowledgements.* This work was supported by a NASA Heliophysics Technology and Instrument Development for Science grant (NNX16AH47G) and a NASA Small Spacecraft Technology Program grant (NNX16AT35A).